# Economic evaluations of interventional opportunities for the management of mental–physical multimorbidity: a systematic review

Amrit Banstola ![ORCID],[1] Subhash Pokhrel ![ORCID],[1] Benedict Hayhoe ![ORCID],[2] Dasha Nicholls ![ORCID],[3] Matthew Harris ![ORCID],[2] Nana Anokye ![ORCID][1]

[1]Department of Health Sciences, Brunel University London, Uxbridge, UK
[2]Department of Primary Care and Public Health, Imperial College London School of Public Health, London, UK
[3]Department of Brain Sciences, Imperial College London Faculty of Medicine, London, UK

**Correspondence to**
Amrit Banstola;
amrit.banstola@brunel.ac.uk

## ABSTRACT

**Objectives** Economic evaluations of interventions for people with mental–physical multimorbidity, including a depressive disorder, are sparse. This study examines whether such interventions in adults are cost-effective.
**Design** A systematic review.
**Data sources** MEDLINE, CINAHL Plus, PsycINFO, Cochrane CENTRAL, Scopus, Web of Science and NHS EED databases were searched until 5 March 2022.
**Eligibility criteria** We included studies involving people aged ≥18 with two or more chronic conditions (one being a depressive disorder). Economic evaluation studies that compared costs and outcomes of interventions were included, and those that assessed only costs or effects were excluded.
**Data extraction and synthesis** Two authors independently assessed risk of bias in included studies using recommended checklists. A narrative analysis of the characteristics and results by type of intervention and levels of healthcare provision was conducted.
**Results** A total of 19 studies, all undertaken in high-income countries, met inclusion criteria. Four intervention types were reported: collaborative care, self-management, telephone-based and antidepressant treatment. Most (14 of 19) interventions were implemented at the organisational level and were potentially cost-effective, particularly, the collaborative care for people with depressive disorder and diabetes, comorbid major depression and cancer and depression and multiple long-term conditions. Cost-effectiveness ranged from £206 per quality-adjusted life year (QALY) for collaborative care programmes for older adults with diabetes and depression at primary care clinics (USA) to £79 723 per QALY for combining collaborative care with improved opportunistic screening for adults with depressive disorder and diabetes (England). Conclusions on cost-effectiveness were constrained by methodological aspects of the included studies: choice of perspectives, time horizon and costing methods.
**Conclusions** Economic evaluations of interventions to manage multimorbidity with a depressive disorder are non-existent in low-income and middle-income countries. The design and reporting of future economic evaluations must improve to provide robust conclusions.
**PROSPERO registration number** CRD42022302036.

## STRENGTHS AND LIMITATIONS OF THIS STUDY

⇒ This systematic review provides a comprehensive review of the cost-effectiveness of interventions seeking to manage multiple long-term conditions, including a depressive disorder in adults.
⇒ In addition to using all major electronic databases, and validated search filters, we judged the economic evidence of each of the included studies based on the checklist in terms of minor, potentially serious and very serious limitations to provide an overall assessment of the review.
⇒ Though we used the recommended checklists to appraise the methodological and reporting quality, they only examined the quality as reported in the studies.
⇒ A network meta-analysis or other quantitative synthesis was infeasible due to methodological and reporting heterogeneity in the included studies.

## INTRODUCTION

Multimorbidity, defined as the presence of two or more long-term conditions in one person, is increasing globally.[1] It affects all ages, but burden is highest among older adults and is associated with increased mortality[2] and reduced health-related quality of life.[3 4] People living with multimorbidity also have functional impairment,[5] higher healthcare utilisation but less continuity of care[6] and pose a significant economic burden to families, health systems and society.[7–10]

Regarding multiple potential combinations of conditions,[11] an area of particular importance is that of mental disorders (eg, depression, anxiety, dementia) and physical disorders (eg, diabetes, cardiovascular disease, arthritis, chronic obstructive pulmonary disease, cancer) in a single individual.[12–14] Mental disorders that accompany long-term physical health conditions exacerbate multimorbidity and associated burden.[15–17] The risk of depression is three times greater in

people with multimorbidity than those without chronic physical conditions,[18] and multimorbidity is more prevalent among individuals with mental disorders (19–21) and those with lower socioeconomic status.[19]

Healthcare services often focus on managing single health conditions and lack coordination across service providers. Such fragmentation is a barrier to effective management of multimorbidity and makes care less likely to be cost-effective.[7] Cost-effective long-term management of multimorbidity is a huge challenge for health systems, patients, health professionals and the community as well as for healthcare decision-makers within resource-constrained settings.[20] Economic evaluation of the prevention and management of multimorbidity is one of the top research priorities acknowledged by the UK Academy of Medical Sciences (AMS), the National Institute for Health and Care Excellence (NICE) and the James Lind Alliance Priority Setting Partnership.[21 22] There is emerging evidence on interventions' effectiveness[23] and cost-effectiveness in tackling multimorbidity in general.[23–25] A recent systematic review included the findings of the economic analysis of two randomised controlled trials (RCTs) for people living with multimorbidity in primary care and community settings[26] targeted interventions such as treatment for depression had shown the potential to be more effective.

Economic evidence of interventions for managing people with mental–physical multimorbidity that includes a depressive disorder is sparse. A recent systematic review identified 11 studies, but none covered mental–physical multimorbidity,[27] and the quality of included studies was reported as poor. Based on current literature, it is unclear whether interventional opportunities to manage mental–physical multimorbidity are cost-effective. This study, therefore, aimed to establish whether interventions seeking to manage multiple long-term conditions, including a depressive disorder in adults, are cost-effective by systematically identifying, collating, reviewing, appraising and summarising the economic evidence. The secondary aim was to critically appraise the methodological quality of the economic evidence.

## METHODS
We used the Preferred Reporting Items for Systematic Reviews and Meta-Analyses (PRISMA) checklist when writing this systematic review.[28] The PRISMA checklist is available in online supplemental file 1. The review protocol was registered in the international prospective register of systematic reviews (PROSPERO) database.[29]

This study adopted a systematic review design with the following attributes (inclusion and exclusion criteria):

### Types of studies
We considered full economic evaluation studies (cost-effectiveness analyses, cost-utility analyses, cost-benefit analyses) conducted alongside randomised, quasi-randomised and non-RCT, modelling studies, controlled

before–after studies and those based on observational studies or analysis of administrative databases that were peer reviewed. Studies conducted in any setting and location were included.

### Types of participants
We defined multimorbidity as coexistence of two or more chronic conditions in the same individual.[22] We included patients age ≥18 years with two or more chronic conditions, of which at least one condition was a depressive disorder (depression, major depressive disorder, persistent depressive disorder or dysthymia) in the same individual.

### Types of interventions
We categorised interventions using the AMS healthcare models for treating patients with multimorbidity.[22] Interventions included any strategy for preventing and treating mental–physical multimorbidity at all healthcare levels. Where interventions had multiple components, we identified the predominant element of the intervention and then categorised them depending on whether they had a predominantly patient or organisational focus:
1. Patient-level interventions:
   Interventions targeted mainly at individuals, for example, educational support and self-management intervention. Such interventions encourage patient self-management and facilitate discussions about personal preferences and priorities with healthcare professionals.
2. Organisational-level interventions and healthcare reform:
   This includes organisational-level changes or changes to the organisation of care. For example, it could be service integration or the provision of coordinated care by multidisciplinary teams (including nurses, physicians and psychiatrists).

### Types of outcome measures
We considered various outcome measures used in economic evaluations, and included, for example, incremental cost-effectiveness ratios (ICERs), cost per depression-free days (DFDs) and treatment success rate.

### Exclusion criteria
► Studies that assessed intervention(s) but did not provide a comparative cost-outcome analysis (ie, cost descriptions/analyses).
► Review articles/literature reviews, systematic reviews, case studies/case reports, study protocols, conference proceedings, opinion pieces (perspective, viewpoint), editorials, letters, commentaries, debates, books, dissertations/theses and abstracts only.

### Search methods for identification of studies
#### Electronic searches
We searched seven electronic databases without restriction on language up until 5 March 2022: (1) MEDLINE, (2) CINAHL Plus, (3) PsycINFO, (4) Cochrane Library,

(5) Scopus, (6) Web of Science and (7) NHS Economic Evaluation Database.

## Search strategy

Existing search strategies were adapted to search for potential studies on 'multimorbidity'[26 30] and 'depressive disorder'.[31 32] In addition, a search filter designed by the Centre for Reviews and Dissemination was used to search potential 'economic evaluation' studies.[33] The search strategy was first designed for MEDLINE and later adapted for other databases. Where there was no existing search filter for a database, the existing search strategies were adapted. The search strategies for each database are provided in online supplemental file 2.

## Searching other resources

We manually searched reference lists of all included studies. In addition, we searched key Cochrane review.[26] Nine of the 17 RCTs included in the review were focused on mental health, particularly depression in people with comorbidities. We checked these nine RCTs (which reported effectiveness) through their trial registries to see whether they had reported cost-effectiveness analysis findings.

## Data collection and analysis
### Selection of studies

All studies identified were exported to EndNote V.X9, and duplicates were removed. Title and abstract of the remaining studies were independently screened by two authors (AB and NA). We retrieved the full text of all studies identified as potentially relevant and assessed each for inclusion. Any disagreement was resolved through discussion and consensus. We excluded studies that did not meet inclusion criteria with the reason for exclusion.

## Data extraction and management

Extraction of all relevant data from included studies was conducted independently by two authors (AB and NA). Any uncertainty was resolved through discussion and consensus. Further information regarding the included studies was retrieved from their associated studies, such as the protocol whenever it was stated as additional sources. We developed a data extraction sheet in Microsoft Excel using an adapted version of the data collection checklists.[34–36]

## Risk of bias assessment in included studies

Critical appraisal of the methodological quality of included studies was undertaken to address risk of bias.[36] The methodological quality of each included study was critically assessed using checklists appropriate to the study's analytical approach by two review authors (AB and NA). Uncertainty was resolved through discussion and consensus. For example, Philips *et al*'s[37] checklist was used to appraise the methodological quality of model-based economic evaluations; Drummond *et al*'s[38] checklist was used to appraise trial-based and other economic evaluations. Quality was used to aid the interpretation of the analysis, not to determine exclusion. The Consolidated Health Economic Evaluation Reporting Standards (CHEERS) checklist was applied to assess quality of the reporting of economic evaluations.[39] Studies were not excluded based on risk of bias assessment.

## Data synthesis and analysis

We adapted the 'economic evidence profile' table from NICE guidance to summarise and present results for economic evaluations of included studies.[40] This table included the following: study details, study limitations (authors' judgement based on the study quality to assess whether it would likely change the results and conclusions), any comments that are helpful to summarise the evidence, price year, incremental costs, incremental effects (eg, quality-adjusted life years (QALYs)), ICER and assessment of uncertainty. Study limitations were categorised as: (a) minor limitations—study meets all quality criteria or fails to meet one or more quality criteria, but this is unlikely to change the conclusions about cost-effectiveness; (b) potentially serious limitations—study fails to meet one or more quality criteria, and this could change the conclusions about cost-effectiveness; (c) very serious limitations—study fails to meet one or more quality criteria, and this is highly likely to change the conclusions about cost-effectiveness. Such studies would usually be excluded from the review.

All costs were converted to 2022 UK Pounds by applying the gross domestic product deflator index and purchasing power parities conversion rate to compare the costs and incremental cost-effectiveness analysis using the Campbell and Cochrane Economics Methods Group (CCEMG)—Evidence for Policy and Practice Information and Coordinating Centre Cost Converter V.1.6.[41]

We included a narrative analysis of the main characteristics and results of included studies. In addition, we presented the results according to the types of intervention and based on the levels of healthcare provision, that is, patient level and organisation level.[22] A network meta-analysis or other quantitative synthesis was infeasible due to methodological and reporting heterogeneity in the included studies.

## Patient and public involvement

Patients or the public were not involved in the design, or conduct, or reporting or dissemination plans of this study.

## RESULTS
### Description of studies

Electronic searches identified 8149 records (including three records identified from citation searching) (figure 1). Of these, 8125 were excluded based on title/abstract review. Full texts were retrieved for 24 studies, of which 19 were considered to have met the inclusion criteria (online supplemental file 3 and 4).

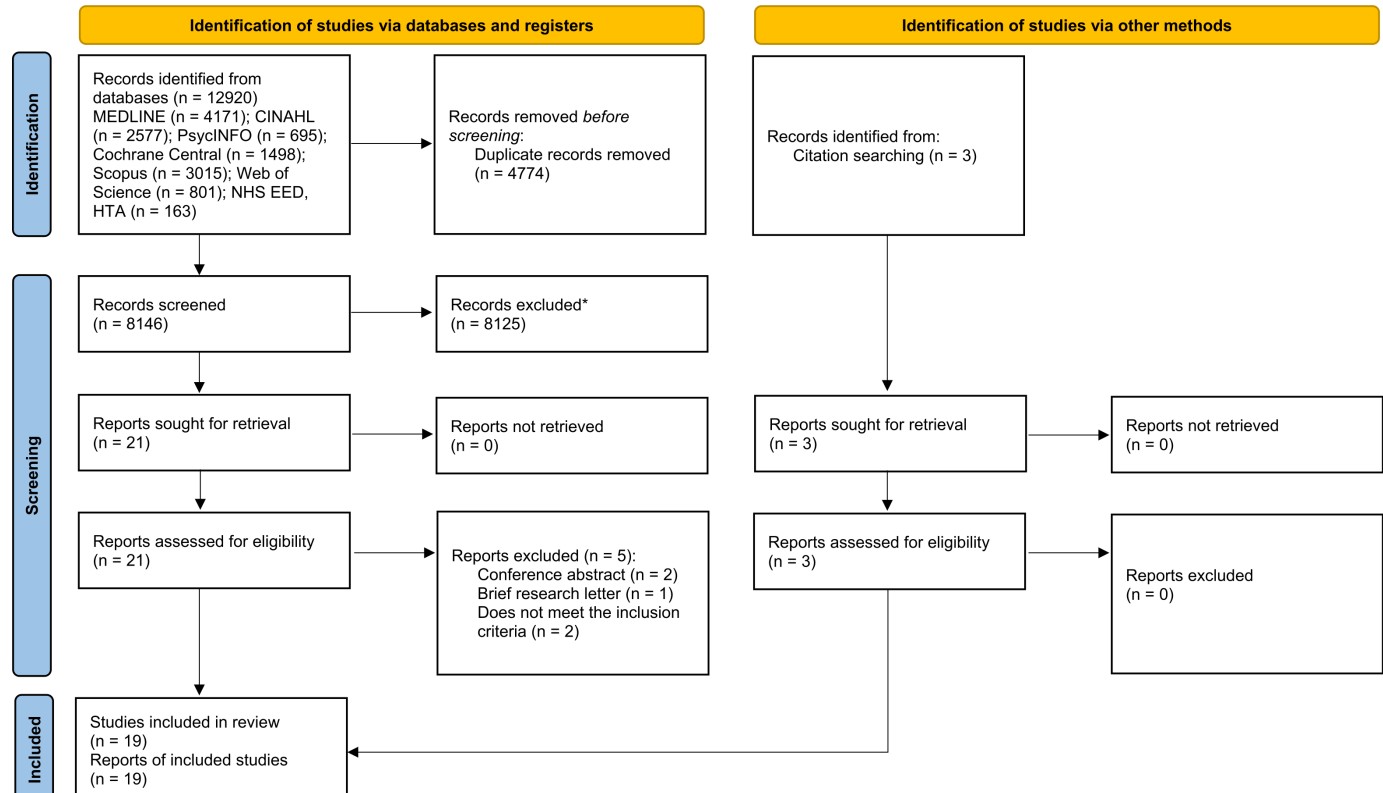

**Figure 1** PRISMA flow diagram. PRISMA, Preferred Reporting Items for Systematic Reviews and Meta-Analyses.

## Study design

Fourteen studies were trials (13 RCTs[42–54] and 1 controlled implementation trial),[55] three were modelling studies,[56–58] one observational (administrative database) study[59] and one pre–post longitudinal study.[60] Eleven studies were cost-utility analyses.[43 48–53 56–58 60] However, only one study was a cost-effectiveness analysis,[47] while seven studies included cost-utility and cost-effectiveness analyses.[42 44–46 54 55 59]

## Quality of included studies

The findings of the assessment of the methodological quality assessment of three model-based studies are presented in online supplemental file 5 and 6 and other studies are presented in online supplemental file 6. The results of the assessment of reporting quality of all studies are presented in online supplemental file 7. The findings of the assessment of both methodological and reporting quality findings showed that there is a great deal of heterogeneity across the studies, as summarised below.

## Study population

Five of the 19 studies recruited patients with a broad range of conditions,[45 49 52 54 56] whereas the remaining 14 focused on the following comorbidities: depression and chronic pain,[42] depression and coronary heart disease,[50] depression and at least one chronic health condition (which is unclear),[60] depression and chronic obstructive pulmonary disease,[53] depression and cardiovascular disease,[59] depression and cancer,[48 51 58] depression and diabetes.[43 44 46 47 55 57]

## Study settings

All 19 studies were undertaken in high-income countries (UK=7,[48–51 56–58] USA=5,[43–45 47 60] Netherlands=2,[52 54] one each in Australia,[53] Canada,[55] Germany,[46] Spain[42] and Taiwan[59]). Ten of 19 studies were set in primary care; four in the UK,[49 50 56 57] three in the USA[44 45 47] and one each in Canada,[55] Spain[42] and Netherlands.[54] Three studies were in UK cancer centres,[48 51 58] two in hospitals (Netherlands[52] and Australia),[53] one in community clinics in the USA[43] and three in other settings (USA,[60] Taiwan[59] and Germany).[46]

## Comparators

The comparator was usual or standard care in most studies. Some studies were supplemented by placebo-befriending phone calls[53] or enhanced care.[43] One study compared the intervention with no intervention or doing nothing scenario[60] and with web-based psychoeducation.[46] One study compared three antidepressants.[59] Two studies had two or more comparators, one being the usual care.[55 57]

## Interventions

Included studies reported four types of interventions. Most were collaborative care[42–45 47–49 51 52 55 56] which in some studies was supplemented by improving rates of opportunistic screening for depression[57] or systematic case identification of depression.[58] Collaborative care in these studies has variable descriptions. However, the main components included case management, follow-up support and coordinated care by multidisciplinary teams of healthcare professionals such as nurses, psychiatrists

and physicians. Other types of interventions include self-management support intervention,[46 50 54 60] telephone-based cognitive behavioural therapy (TB-CBT)[53] and antidepressant treatment.[59] A detailed description of each intervention for each study is provided in online supplemental file 8.

In five studies,[46 53 54 59 60] the interventions were primarily patient-focused, for example, self-management. In the remaining 14 studies,[42–45 47–52 55–58] the interventions identified had a predominantly organisational focus (eg, multidisciplinary teams of healthcare professionals), although some comprised patient-level elements, for example, case management.

### Key design aspects
Other key design aspects of the included studies in relation to perspectives taken; time horizon and discount rates used; selection, measurement and valuation of outcomes; costing approaches; handling of uncertainty and health economic analysis plans are described in online supplemental file 9. In summary, the included studies varied hugely in the way they applied or reported on these design aspects.

### Cost-effectiveness results
Three studies had very serious limitations[53 59 60] largely due to the study design that showed evidence of the effectiveness. The study design was an observational study based on an administrative database[59] or pre–post longitudinal design.[60] Although one study was an RCT, the study duration was inadequate (only 17 weeks) to capture all relevant costs and outcomes.[53] Nine studies had potentially serious limitations.[43–46 48 51 52 55 56] These studies were judged as potentially serious limitations for reasons such as using non-validated measures to estimate QALY[44 45] and duration of the trial less than a year.[46 48 51] There was also a statistically significant imbalance between study groups at baseline randomisation,[43] or no randomisation of comparison groups in a trial,[55] relatively small sample size[52] and extrapolation of short-term (4 month) trial data to estimate cost-effectiveness.[56] The remaining seven studies had minor limitations[42 47 49 50 54 57 58] as sensitivity analysis was conducted to only a few parameters whose values were uncertain, but this was unlikely to change the conclusions about cost-effectiveness (online supplemental file 10).

Cost-effectiveness by levels of healthcare provision and type of interventions are presented in online supplemental file 10 and are summarised briefly below.

### Patient-level interventions (five studies)
#### Self-management
Three of the five patient-level interventions were self-management support interventions[46 54 60]; however, they were focused on different disease clusters. In Germany, a cost-effectiveness analysis alongside an RCT found that GET.ON Mood Enhancer Diabetes (GET.ON M.E.D.) (a web-based self-management support intervention) compared with web-based psychoeducation had an ICER of £11 274 per QALY gained and £245 per treatment response in adults with comorbid depression and diabetes.[46] However, this analysis was assessed as having potentially serious limitations. Cost-effectiveness analysis alongside an RCT found that Minimal Psychological Intervention (a self-management support based on cognitive behavioural therapy) was dominant (the intervention was less costly but more effective) compared with usual care for older adults with multiple long-term conditions in the Netherlands.[54] This analysis was assessed as having minor limitations. A cost-utility analysis based on a pre–post longitudinal study found that the 'Chronic Disease Self-Management Programme' compared with 'no intervention' had an ICER of £31 540 per QALY gained in adults with depression and at least one chronic health condition in the USA.[60] However, this analysis was assessed as having very serious limitations.

#### Telephone-based cognitive behavioural therapy
In Australia, a cost-utility analysis alongside RCT found that TB-CBT compared with standard care plus placebo-befriending phone calls had an ICER of £27 958 per QALY gained in adults with depression and anxiety comorbidities with chronic obstructive pulmonary disease.[53] However, this study was assessed as having very serious limitations.

#### Antidepressant treatment
One analysis based on the national health insurance research database record that compared three antidepressants treatment found that selective serotonin reuptake inhibitors (SSRIs) antidepressant treatment was dominant compared with serotonin norepinephrine reuptake inhibitors.[59] SSRIs compared with tricyclic antidepressants were considered cost-effective by the authors (£55 per percentage point of treatment success) and had an ICER of £55 394 per QALY gained for adults with comorbid cardiovascular disease and depression in Taiwan.[59] However, this analysis was assessed as having very serious limitations.

### Organisational-level interventions (14 studies)
#### Collaborative care for people with depressive disorder and diabetes
Five studies (three from the USA,[43 44 47] one from Canada[55] and another from the UK)[57] reported the cost-effectiveness of collaborative care for people with depressive disorder and diabetes. Cost-utility analysis alongside RCT also from the USA found that the 'Multifaceted Diabetes and Depression Programme' compared with 'enhanced usual care' had an ICER of £3543 per QALY gained for low-income Hispanic adult patients.[43] This analysis was assessed as having potentially serious limitations. Cost-effectiveness analysis alongside RCT found that 'IMPACT intervention' compared with usual care had an ICER of £206 to £413 per QALY gained and less than £1 per DFDs for elderly patients at primary care clinics in

the USA.[44] This analysis was assessed as having potentially serious limitations. Another analysis alongside RCT from the USA found that the 'systematic depression treatment programme' was dominant compared with usual care among outpatients of middle-aged to elderly patients.[47] This analysis was assessed as having minor limitations.

In Canada, a cost-effective analysis alongside RCT found that collaborative care compared with enhanced had £7 per DFDs and an ICER of £10 803 per QALY gained. Compared with usual had £6 per DFDs and an ICER of £16 597 per QALY gained care for adult patients.[55] This analysis was assessed as having potentially serious limitations. In England, a model-based cost-utility analysis found that policy changes (that include collaborative care) to improve the current care pathway was cost-effective (£12 656 per QALY gained; decision threshold £20 000/ QALY) compared with current practice in adults.[57] This analysis was assessed as having minor limitations.

### Collaborative care for people with comorbid major depression and cancer

Three studies from the UK reported the cost-effectiveness of collaborative care intervention 'Depression Care for People with Cancer (DCPC)' for people with comorbid major depression and cancer.[48 51 58] An earlier cost-utility analysis alongside RCT found that the DCPC was potentially cost-effective (£7098 per QALY gained; decision threshold £20 000/QALY) compared with usual care in adults attending specialist medical services in Scotland.[48] Another cost-utility analysis alongside multicentre RCT found that the DCPC was cost-effective (£11 802 per QALY gained) compared with usual care for adult patients in Scotland.[51] The probability of the intervention being cost-effective was over 90% at the current threshold of £20 000 per QALY. Both these analyses were assessed as having potentially serious limitations. A model-based cost-utility analysis found that the 'systematic integrated depression management' (that includes DCPC) was cost-effective (£14 540 per QALY gained) compared with usual practice for adult patients.[58] The probability of the DCPC being cost-effective in this study was over 99% at a threshold of £20 000 per QALY. This analysis was assessed as having minor limitations.

### Collaborative care for people with depression and multiple long-term conditions

Four studies (one each from the USA[45] and the Netherlands,[52] two from the UK)[49 56] reported the cost-effectiveness of collaborative care intervention for people with depression and multiple long-term conditions. Cost-effectiveness analysis alongside RCT found that the collaborative treatment programme 'TEAMcare' was dominant compared with the usual primary care in outpatients for adult patients in the USA.[45] The probability that the intervention would be cost-effective was 99.7% based on a threshold of US$20 000 per QALY. This analysis was assessed as having potentially serious limitations. In England, a model-based cost-utility analysis conducted

during an RCT (at 4 months) found that collaborative care could be cost-effective (£18 580 per QALY gained) compared with usual care for adult patients.[56] The probability of the intervention being cost-effective was 53% at the threshold of £20 000 per QALY. Subsequent cost-utility analysis at the end of the RCT (at 2 years) reported a lower cost (£14 995) per additional QALY gained from collaborative care with 75% and 92% probability of being cost-effective at the threshold of £20 000 and £30 000 per QALY, respectively.[49] Both these analyses were assessed as having minor limitations. In the Netherlands, cost-utility analysis alongside multicentre RCT found that collaborative care compared with usual care had an ICER of £27 674 per QALY gained from a healthcare perspective and an ICER of £24 088 per QALY gained from a societal perspective for adult patients.[52] This analysis was assessed as having potentially serious limitations.

### Collaborative care for people with major depression and chronic musculoskeletal pain

Cost-effectiveness analysis alongside RCT found that collaborative care intervention 'DepRessiOn and Pain' compared with usual care had an ICER of £28 495 per QALY gained from a healthcare system perspective and an ICER of £28 629 per QALY gained from a societal perspective for adults with major depression and chronic musculoskeletal pain in Spain.[42] The DFDs from both the healthcare system and societal perspective were £34 per DFDs. This analysis was assessed as having minor limitations.

### Self-management (personalised care for people with depression and coronary heart disease)

In England, a cost-utility analysis alongside a multicentre RCT pilot study found that personalised care intervention 'UPBEAT' was not cost-effective (£36 979 per QALY gained; decision threshold £20 000/QALY) compared with treatment as usual for adult patients with depression and coronary heart disease.[50] However, the authors claimed that it has the potential to be more cost-effective up to a threshold of £3035 per QALY. This analysis was assessed as having minor limitations.

### DISCUSSION

To the best of our knowledge, this is the most comprehensive review of the literature on economic evidence around interventional opportunities for managing mental–physical multimorbidity. While there is evidence of potentially cost-effective interventions in high-income countries (HICs), no study has been found to reflect the cost-effectiveness of mental–physical multimorbidity management in low and middle-income countries (LMICs). A question, therefore, arises whether (and to what extent) the HICs evidence in this area would be transferable to LMICs. Before attempting to answer this question, it is important to discuss the wider implications of our findings first.

Both patient-level and organisational-level interventions have been found to be potentially cost-effective. Patient-level interventions, such as self-management support intervention in multiple long-term conditions and interventions that target comorbid depression and diabetes, could be more cost-effective compared with usual care. Organisational-level intervention, particularly collaborative care, is more likely to be cost-effective compared with usual care. Therefore, both HICs and LMICs can consider designing and implementing interventions to manage mental–physical multimorbidity at both individual and organisational levels to ensure that they get the best return on their investment in this area.

In the UK, existing NICE guidelines recommend using collaborative care only for patients with moderate to severe depression alongside other comorbid long-term physical health conditions such as cancer, heart disease or diabetes.[61] While organisational interventions, particularly collaborative care for people with depressive disorder and diabetes, comorbid major depression and cancer, and depression and multiple long-term conditions, could be cost-effective, collaborative care for people with major depression and chronic musculoskeletal pain, TB-CBT for people with depression and chronic obstructive pulmonary disease and personalised care intervention 'UPBEAT' for people with depression and coronary heart disease were not cost-effective. This highlights how complex interventional opportunities for multimorbidity management can be. For example, the cost-effectiveness of organisational-level interventions such as collaborative care can vary depending on how psychological morbidities interact with certain types of physical morbidities.

There is no consensus regarding the definition of multimorbidity,[1 62] which makes comparison of studies challenging. The AMS definition of multimorbidity includes a physical non-communicable disease of long duration, such as cardiovascular disease or cancer; a mental health condition of long duration, such as a mood disorder or dementia and an infectious disease of long duration, such as HIV or Hepatitis C.[22] The NICE definition of multimorbidity includes any defined physical or mental health conditions, such as diabetes or schizophrenia; ongoing conditions, such as learning disability; symptom complexes, such as frailty or chronic pain; sensory impairment, such as sight or hearing loss and alcohol or substance misuse among others.[63] Furthermore, although the term multimorbidity has been used in health research since 1976,[64] it was only 20 years later that the distinction between multimorbidity and comorbidity was recognised.[65] Multimorbidity was recognised as the Medical Subject Headings in early 2018. Before that, comorbidity was more common and used interchangeably.[66] Therefore, the cost-effectiveness implications reported in this systematic review should not be taken as 'blanket evidence' as they are valid only for the types of multimorbidity and their management that have been contextualised by individual studies. When taken to LMICs, such contextualisation (of target populations,

interventions, comparators and outcomes) remains even more important to consider in any future design and evaluation of interventional opportunities to manage mental–physical multimorbidity.

Our attempt to report studies from different countries and currencies in the UK Pound may facilitate a degree of direct comparison of the cost-effectiveness of different interventions but it does not suggest these interventions are transferable across jurisdictions.[67] The transferability (both applicability and generalisability) of the findings obtained from these studies to another setting, therefore needs to be assessed. There are always variations in patient population composition, the healthcare delivery system, healthcare financing and unique socioeconomic conditions across jurisdictions. For example, unlike in HICs, multimorbidity is more prevalent in people with higher socioeconomic status than those with lower socioeconomic status in countries such as India, Ghana and Russia.[22] The findings from this study could help HICs and LMICs to look for both individual and organisational-level opportunities to intervene, but such interventions must be designed and implemented to maximise their cost-effectiveness through appropriate contextualisation as described above. Although there is a relatively better understanding and choice on assessing outcomes using either QALYs or disability-adjusted life years, for the costs, it is often unclear which cost items to include. To facilitate consistency and improve study comparability, studies should consider including direct medical care use costs (interventions, treatment, medication, laboratory and diagnostic services, primary and secondary care, hospital inpatient and outpatient care, emergency department visits, different healthcare professionals consultation, workshop sessions, training); direct non-medical care use costs (travel to healthcare appointments, informal care) and indirect costs (productivity loss). Researchers can include other items relevant to local context and study purposes.

### Quality of the evidence and guidance for addressing methodological challenges

Methodological and reporting heterogeneity found across the included studies meant that a quantitative analysis of the findings to generate an 'average' cost-effectiveness figure for a specific type of intervention was not feasible. There are numerous economic evaluation guidelines, but they all seem to overlap in part and share similarities.[68] We, therefore, felt that there is no need for separate guidance on this topic as the existing available guidelines on economic evaluation, if used appropriately, are still applicable and relevant. We strongly recommend that the future economic evaluation study in this area follows the established economic evaluation checklists such as Drummond Checklist,[38] Consensus Health Economic Criteria (CHEC) list,[69] Phillips checklist (for model-based economic evaluation)[37] and updated CHEERS 2022 checklist to report the economic evaluation evidence.[39] For those devising a systematic review of

economic evaluation on this topic, we recommend the recent version of the Cochrane Handbook for Systematic Reviews of Interventions[70] supplemented by 'Chapter 15: Incorporating economics evidence' of earlier V.5.1.0.[36] Other valuable resources included guidance from the Centre for Reviews and Dissemination of the University of York,[35] the NICE[40] and the Joanna Briggs Institute[71] among others. A slight adaptation to these existing guidelines may suffice should the complexity of this topic rises in the future, particularly around contextualisation of the intervention.

### Strengths and limitations

The justified choices made in the design and implementation of this study have improved transparency, comprehensiveness and replicability of this systematic review that has identified—possibly for the first time—a number of cost-effective interventional opportunities to manage mental–physical multimorbidity at both individual and organisational levels. One of the major limitations of this study is the exclusion of grey literature, unpublished evaluation and no provision to contact experts or authors of the published paper. This could have led to an 'omission bias'. Though we used the recommended checklists to appraise the methodological and reporting quality, they only examined the quality as reported in the studies. Assessment of the risk of bias of the main studies on which economic evaluations were based (eg, RCTs) was beyond the scope of this study.

### Implications for practice and policy

This review suggests that organisational interventions, particularly collaborative care for people with depressive disorder and diabetes, comorbid major depression and cancer and depression and multiple long-term conditions, could be cost-effective in improving the management of mental–physical multimorbidity. Policymakers should prioritise such interventions for implementation in order to optimise resource allocation. There may be a need for targeted government funding and support programmes to implement this programme as it demands modification of the current clinical practices, which mostly rely on a single-disease treatment approach. This is particularly appropriate as the number of people with mental–physical multimorbidity is projected to increase, and concern over the ability of an already resource-constrained healthcare system, particularly in LMICs.

### Implications for future research

Future economic evaluations in this area must improve both in design and reporting to minimise risk of bias. In addition, future economic evaluations should examine distributional cost-effectiveness to understand better the equity aspects of implementing cost-effective interventions to address mental–physical multimorbidity.[72] There is a need for further economic evaluation studies of various potential disease clusters primarily from LMICs and based in both primary care and community settings.

If designing RCTs of the interventions to manage mental–physical multimorbidity, future research needs to examine trial-based and model long-term cost-effectiveness of the interventions. Where appropriate, future studies could include other non-health benefits such as improved productivity, reduced absenteeism and decreased family burden for care to increase the evidence base on this important area.

## CONCLUSION

The economic evidence on the interventions to manage multiple long-term conditions with a depressive disorder is limited to HICs. Organisational interventions, particularly collaborative care for people with depressive disorder and diabetes, comorbid major depression and cancer and depression and multiple long-term conditions, seem more likely to be cost-effective. LMICs can use this knowledge base to design their own interventions to manage mental–physical multimorbidity, paying special attention to contextualisation of specific interventions.

**Correction notice** This article has been corrected since it first published. The guarantor's name has been updated from Amrit Banstola (AB) to Nana Anokye (NA) in the contributor statement.

**Acknowledgements** We acknowledge Brunel University London library for their support in identifying research articles.

**Contributors** AB and NA contributed to the conception and design of this research. AB performed the literature search, and AB and NA reviewed articles, extracted data from individual studies, conducted the data analysis and interpreted the data. AB wrote the first draft of the paper, which was commented on by SP, BH, DN, MH and NA. All authors contributed to the interpretation of the findings, revised the manuscript for important intellectual content and agreed to the final draft of the manuscript. NA is responsible for the overall content as the guarantor.

**Funding** This article presents independent research commissioned by the UK National Institute for Health Research (NIHR) under the Applied Research Collaboration (ARC) programme for Northwest London.

**Disclaimer** The views expressed in this publication are those of the author(s) and not necessarily those of the UK National Health Service, the NIHR or the UK Department of Health and Social Care.

**Competing interests** None declared.

**Patient and public involvement** Patients and/or the public were not involved in the design, or conduct, or reporting, or dissemination plans of this research.

**Patient consent for publication** Not applicable.

**Ethics approval** Not applicable.

**Provenance and peer review** Not commissioned; externally peer reviewed.

**Data availability statement** All data relevant to the study are included in the article or uploaded as supplementary information.

**ORCID iDs**
Amrit Banstola http://orcid.org/0000-0003-3185-9638
Subhash Pokhrel http://orcid.org/0000-0002-1009-8553
Benedict Hayhoe http://orcid.org/0000-0002-2645-6191
Dasha Nicholls http://orcid.org/0000-0001-7257-6605
Matthew Harris http://orcid.org/0000-0002-0005-9710
Nana Anokye http://orcid.org/0000-0003-3615-344X

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
