## [Reviewer comments · BMJ Open]

ARTICLE DETAILS

TITLE (PROVISIONAL)	Economic evaluations of interventional opportunities for the management of mental-physical multimorbidity: a systematic review
AUTHORS	Banstola, Amrit; Pokhrel, Subhash; Hayhoe, Benedict; Nicholls, Dasha; Harris, Matthew; Anokye, Nana

VERSION 1 – REVIEW

REVIEWER	Gupta, Priti Centre for Chronic Disease Control
REVIEW RETURNED	14-Nov-2022

GENERAL COMMENTS	Paper is very well written but the section where authors have mentioned about quality of studies included in this review they have simply mentioned studies with major or minor limitation, this would be better if they can mention briefly about what these limitations were.
---

REVIEWER	Gc, Vijay University of Huddersfield School of Human and Health Sciences
REVIEW RETURNED	06-Jan-2023

GENERAL COMMENTS	This is a well-designed, well-structured review paper that explores the cost-effectiveness of interventions seeking to manage multiple long-term conditions, including depressive disorder in adults. The manuscript does an excellent job of noting shortcomings in the methodological quality and transferability of the findings to LMICs. Although more details are provided in the supplementary files, it would be useful to briefly discuss the range of costs included in economic evaluations and what better (or minimal) ones would be. Overall, however, I found it to be an excellent paper that I expect to be very useful to researchers conducting economic evaluations in this area in future.
---

VERSION 1 – AUTHOR RESPONSE

Reviewer: 1

Comment: Paper is very well written but the section where authors have mentioned about quality of studies included in this review they have simply mentioned studies with major or minor limitation, this would be better if they can mention briefly about what these limitations were.

Response: Thank you for your suggestions. For those studies with 'very serious limitations' and 'potentially serious limitations', we have already stated what the limitations were on page 6 of the manuscript. For the studies with 'minor limitations', we have now amended the manuscript to briefly mention what the limitations were.

“The remaining seven studies had minor limitations as sensitivity analysis was conducted to only a few parameters whose values were uncertain, but this was unlikely to change the conclusions about cost-effectiveness.”

Please see the track changes on page 6 of the revised manuscript.

Reviewer: 2

Comment: This is a well-designed, well-structured review paper that explores the cost-effectiveness of interventions seeking to manage multiple long-term conditions, including depressive disorder in adults. The manuscript does an excellent job of noting shortcomings in the methodological quality and transferability of the findings to LMICs. Although more details are provided in the supplementary files, it would be useful to briefly discuss the range of costs included in economic evaluations and what better (or minimal) ones would be. Overall, however, I found it to be an excellent paper that I expect to be very useful to researchers conducting economic evaluations in this area in future.

Response: Thank you for your suggestions. We have added a few sentences on page 9 to reflect this.

“Although there is a relatively better understanding and choice on assessing outcomes using either quality-adjusted life years or disability-adjusted life years, for the costs, it is often unclear which cost items to include. To facilitate consistency and improve study comparability, studies should consider including direct medical care use costs (interventions, treatment, medication, laboratory and diagnostic services, primary and secondary care, hospital inpatient and outpatient care, emergency department visits, different healthcare professionals consultation, workshop sessions, training); direct non-medical care use costs (travel to healthcare appointments, informal care); and indirect costs (productivity loss). Researchers can include other items relevant to local context and study purposes.”

VERSION 2 – REVIEW

REVIEWER	Gc, Vijay University of Huddersfield School of Human and Health Sciences
REVIEW RETURNED	06-Feb-2023
GENERAL COMMENTS	The authors have addressed the comments in an adequate way.

VERSION 2 – AUTHOR RESPONSE